# Relationship between Organizational Climate and Service Performance in South Korea and China

**Xuezhe Quan [1], Myeong-Cheol Choi [2],\* and Xiao Tan [2],\***

1 Department of Business, Kyunghee University, Seoul 02447, Republic of Korea; qxz920706@gmail.com
2 Department of Business, Gachon University, Seongnam 13120, Republic of Korea
\* Correspondence: oz760921@gachon.ac.kr (M.-C.C.); 202255020@gachon.ac.kr (X.T.)

**Abstract:** Both South Korea and China have collective cultures; however, there are significant differences in employee behavior due to cultural, economic, and environmental factors. This study explores the influence of organizational climate on employee innovative behavior and service performance using a competitive value model, as well as the mediating effects of social capital and organizational silence. Adopting the interpersonal relationship, rational goal, and internal process approaches, it focuses on three aspects: supervisory support, pressure to produce, and formalization. A total of 773 valid questionnaires were collected from four- and five-star hotels in South Korea and China, and the data were analyzed using SPSS and AMOS. The results showed that supervisory support and pressure to produce positively affected employee social capital, thereby affecting their service performance. Formalization positively affected organizational silence and negatively affected employees' innovative behaviors. This study confirmed the mediating effects of social capital and organizational silence in the organizational environment. The positive effects of supervisory support and pressure to produce on social capital were similar in South Korea and China. However, among the effects of organizational silence, Korean employees were more likely to benefit from formalization. This study identified the differences in organizational climate and organizational performance between South Korea and China and provides implications for enterprises' sustainable development.

**Keywords:** organizational climate; innovative behavior; service performance; social capital; organizational silence

## 1. Introduction

Globalization enables companies to have employees from different cultures working together. In international companies, employees with different cultural backgrounds are affected differently by the organizational climate. Certain organizational climates can positively impact employees from some cultures [1] but negatively impact those from others. Korean and Chinese cultures have been categorized as collectivist, and little research has been conducted on the differences between employees from these cultures [2]. Only a few studies have examined the differences between the impacts of organizational climate on Korean and Chinese employees' performances. Consequently, this study examines how organizational climate affects employee service and innovation. Organizational performance is affected by customer satisfaction [3], service performance, and innovation behavior.

Generally, employee performance is determined by organizational goals and control over individual performance [4]. Innovative behavior refers to the application of new ideas, products, processes, or procedures by employees in their work [5,6], depending on various factors, including personality, willingness to innovate, level of organizational support [7], knowledge sharing, and absorptive capacity [8]. Social capital and organizational silence are important factors in organizational research. Specifically, social capital is highly correlated with employee competency [9] and is seen as a valuable source of productive capacity to deliver better service [10]. It plays an important role in the relationship between organizational climate and service performance [11]. Organizational

silence refers to behavior associated with negative feedback from others and can refer to a situation in which employees intentionally or unintentionally withhold information that could be valuable to their organization [12,13]. Chen et al. [14] emphasized the negative effects of employee silence on organizations. Therefore, organizational silence may affect the relationship between organizational climate and innovation behavior.

There may be differences between the Korean and Chinese cultures regarding the internal mechanisms responsible for the relationships between organizational climate, employee performance, and innovative behavior. Both South Korea and China have Confucian cultures; however, there are some differences in employee behavior that are influenced by history, systems, and national culture. It could be said that all cultures embody collective knowledge [15], and entrepreneurs' behavior and perceptions are essentially shaped by national culture [16]. Despite the fact that Korea and China are widely recognized as having collective cultures, Korea is a capitalist country that is heavily influenced by the United States, whereas China is socialist. Compared with Americans, the Chinese are more accurate at recognizing collective emotions [17]; thus, there may be differences in employees' cultures in Korea and China.

This study examines the impact of organizational climate on employees' performances and innovative behaviors, as well as the mediating effects of social capital and organizational silence. Additionally, it confirms the differences in organizational behavior between Korean and Chinese employees, providing a country-based comparison and an extension for studying organizational climate.

## 2. Theory and Hypothesis Development

Social exchange theory (SET) posits that, when an employer takes care of an employee, the employee repays the organization with a more positive work attitude and behavior [18]. Such interaction with a sense of obligation leads to the formation of a social exchange relationship between the employee and the organization [19]. Social interaction theory holds that social interaction is a process in which people and groups interact with one another within the society [20]. Based on SET and social interaction theory, this study constructed a competitive value model that included organizational climate, service performance, innovation behavior, social capital, and organizational silence.

Organizational climate has been described as the perception of an organization by its employees; however, its construction has been plagued by conflicting definitions and inconsistencies in implementation for many years. The dominant approach describes climate as employees' shared perceptions of an organization's events, practices, and procedures, which are characterized as descriptive rather than emotional or evaluative [21]. At the level of individual analysis, referred to as "psychological climate" [22], these perceptions represent how the work environment is cognitively evaluated and represented in terms of its meaning and importance to an individual employee within an organization [22]. Organizational climate has a positive effect on organizational performance [23].

This study examined the cognitive aspects of organizational climate using the competitive value model as a meta-theoretical framework [24–28] and the basic framework for organizational climate. Therefore, it focused on three aspects of competitive value models: the human relationship, internal process, and rational goal approaches.

The human relationship approach reflects traditions derived from socio-technical [29] and human relations [30] as a means of representing the wellbeing, growth, and commitment of a community of workers within an organization. It is based on an internal focus and flexibility in relation to the environment. The rational goal approach (externally focused, but tightly controlled within an organization) reflects a rational economic model of organizational functioning focused on productivity and goal achievement [31,32]. The internal process approach (internal focus and tight controls within an organization) reflects the formalization of systems and internal controls to ensure the efficient use of resources and to achieve business performance [33].

### 2.1. Organizational Climate, Social Capital, and Organizational Silence

The causal approach in organizational climate refers to supervisory support and helps employees improve their job satisfaction. However, the rational goal approach refers to pressure to produce and helps employees improve their productivity and efficiency. Consequently, both organizational climates can result in employees' greater commitment to their jobs [34]. Social support helps employees remain engaged in their jobs [35]. Job resources, including social support, promote employee engagement at work [36,37]; employees tend to be more engaged at work when they receive support from their employers, supervisors, or coworkers [34]. Pressure to produce occurs when a person experiences work-related mental, physical, or emotional strain through physical and physiological effects [38]. Pressures of the work environment, such as workload and communication, increase employees' fears of resource loss, which in turn, leads to an increase in job engagement in the quest for more resources [39]. High company and job commitment is positively correlated with employee social capital [40]; thus, supervisory support and pressure to produce may play a positive role in employee social capital.

Formalization can be viewed as a strategy to minimize uncertainty within organizational cultures because it represents the internal process model of an organization [41]. East Asian countries have a higher degree of uncertainty avoidance than western countries because they have high-context cultures. Formalization significantly facilitates voice, promoting organizational citizenship behaviors (OCBs) to a lesser extent [42]. The formalization of organizational climate can, therefore, encourage Korean and Chinese employees to avoid uncertainty, resulting in increased organizational silence.

**Hypothesis 1 (H1).** *Supervisory support (a), pressure to produce (b), and organizational climate are positively related to social capital.*

**Hypothesis 2 (H2).** *The formalization of organizational climate is positively related to organizational silence.*

### 2.2. Social Capital, Organizational Silence, Service Performance, and Innovative Behavior

Social capital is an important resource for workplace socialization, which positively impacts employee performance [43–45]. An organization's social capital is a set of informal values, norms, and subjectively felt duties shared by employees. It plays an important role in shaping relationships that allow an organization to operate efficiently [46]. Social capital is a fundamental driver of performance and serves as a catalyst to foster stronger connections within social networks by creating an environment of trust and collaboration [46–48]. Organizational silence is caused by employee desire to avoid negative feedback [12] and affects organizational performance [49]. Negative feedback provides individuals with more useful information than positive feedback [50,51]. A lack of organizational silence negatively impacts innovative behavior because innovative behaviors that require a significant number of new elements are more likely to receive negative feedback.

**Hypothesis 3 (H3).** *Social capital is positively related to service performance.*

**Hypothesis 4 (H4).** *Organizational silence is negatively related to innovative behavior.*

### 2.3. Social Capital and the Mediating Effects of Organizational Silence

Results indicate that supervisory support and an organizational climate with pressure to produce increase employee job commitment [23], which in turn, affects their social capital. Social capital is an important resource for employees' social activities and can lead to improved service quality. Due to formalized organizational climates, east Asian employees with high levels of uncertainty avoidance engage in organizational silence. This

organizational silence may impede the implementation of innovative behaviors that may encounter negative feedback.

Previous research has demonstrated that organizational climate is associated with various important performance factors at the individual level. Brown and Leigh [52] demonstrated that job performance was positively related to organizational climate, which motivated employees. According to Cabrera and Cabrera [53], an organizational environment that provided employees with a sense of security and in which they were not criticized without reason was conducive to innovative thinking. Employees who feel comfortable in an organization are more likely to create and share knowledge [54,55].

**Hypothesis 5 (H5).** *Social capital mediates the relationships among supervisory support (a), pressure to produce (b), organizational climate, and service performance.*

**Hypothesis 6 (H6).** *Organizational silence mediates the relationship between formalization of organizational climate and innovative behavior.*

*2.4. Comparison of Korea and China*

Generally, labor values fall into two categories: intrinsic and extrinsic [56]. The distinction between intrinsic and extrinsic values relates to the source of value of an activity, state of affairs, or object [57]. Intrinsic value is the state of being that occurs through work or as a result of a person's dedication to a job. It is directly linked to job content, such as a sense of accomplishment. Individuals with strong intrinsic work values have stronger intrinsic motivation [58]. In contrast, extrinsic work value refers to the ultimate state of achievement as a result of work, regardless of its content, such as salary satisfaction. Work values have been examined from a cross-cultural perspective in numerous studies. Kim Hee-chul and Kim Jong-rim [59] compared Korean and Chinese workers' work values and found that Korean workers valued organizational management and achievement factors more than Chinese workers, while Chinese workers valued promotion and salary factors more than Korean workers. The effects of different types of work values on creative performance vary [60]. The results of Bang's [61] analysis of Korean and Chinese work values were as follows: In terms of labor perspectives, the Chinese preferred to work according to official regulations, tended to make decisions regarding work individually rather than collectively, and were highly motivated to work overtime. Regarding job attitudes, Koreans placed greater emphasis on work, whereas Chinese workers placed greater emphasis on salaries. Koreans cited pressure to produce as one of their workplace challenges, whereas Chinese employees said that low salaries were among their challenges. It was observed that Korean employees placed greater value on intrinsic work values, whereas Chinese employees placed greater emphasis on extrinsic work values. Employees' preferences for work values profoundly affect work performance [62]. In this regard, Korean employees, who value work processes more than their Chinese counterparts, are more engaged.

Uncertainty avoidance is the cultural value of feeling anxious or threatened in uncertain or ambiguous situations [63]. Existing research suggests that China exhibits less uncertainty aversion than east Asian countries, such as South Korea [64] (Figure 1).

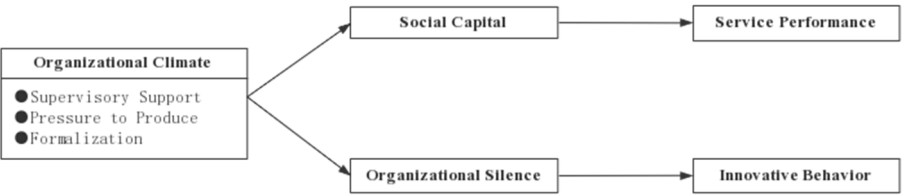

**Figure 1.** Research model.

**Hypothesis 7 (H7).** *The positive relationships among supervisory support (a), pressure to produce (b), organizational climate, and social capital are stronger for Korean employees than for Chinese employees.*

**Hypothesis 8 (H8).** *The positive relationship between a formalized organizational climate and organizational silence is stronger for Korean employees than for Chinese employees.*

## 3. Methodology

### 3.1. Data Collection

Data were collected from July to September 2022 through on-site visits and online surveys of employees at four- and five-star hotels in Seoul and Gyeonggi province, South Korea, and Beijing and Hebei province, China. We used random sampling to collect 773 questionnaires (280 from Korea and 493 from China). The respondents' demographic characteristics are shown in Table 1.

**Table 1.** Demographic characteristics of the sample.

| Demographic Variable | Type | Korea | China |
|---|---|---|---|
| Gender | Male | 130 | 180 |
| | Female | 150 | 313 |
| Age | Younger than 25 years | 49 | 224 |
| | 26–30 years old | 78 | 95 |
| | 31–35 years old | 69 | 79 |
| | 36–40 years old | 39 | 60 |
| | 41–50 years old | 31 | 25 |
| | Older than 50 years | 14 | 10 |
| Educational Background | High school degree or below | 15 | 15 |
| | College degree | 84 | 299 |
| | Bachelor's degree | 161 | 141 |
| | Master's degree | 16 | 26 |
| | PhD degree or above | 4 | 12 |
| Tenure | Work for 0–5 years | 121 | 285 |
| | Work for 6–10 years | 71 | 103 |
| | Work for 11–15 years | 39 | 67 |
| | Work for 16–20 years | 29 | 17 |
| | Over 20 years of work | 20 | 21 |
| Position | Contract Workers | 39 | 254 |
| | Staff | 111 | 93 |
| | Assistant Manager | 73 | 86 |
| | General Manager | 38 | 43 |
| | Department Manager | 12 | 17 |
| | Executive | 4 | 0 |
| | Other | 3 | 0 |
| Type of Job | Planning/Advertising | 111 | 181 |
| | Sales/Marketing | 77 | 170 |
| | Management/Office | 91 | 134 |
| | Other | 1 | 0 |
| Organizational Size | Fewer than 50 people | 15 | 145 |
| | 51–100 people | 52 | 74 |
| | 101–200 people | 36 | 77 |
| | 201–500 people | 75 | 73 |
| | More than 500 people | 102 | 124 |
| Total | | 280 | 493 |

*3.2. Measurement Tools*

3.2.1. Organizational Climate

The independent variable, organizational climate, was operationally defined as "the interaction between an organization's important environmental factors and the values, attitudes, and beliefs of the people working in that particular organization". The items were adapted from Patterson et al. [65] and were measured on a 5-point Likert scale, with 13 items categorized in supervisory support (leader), innovation and flexibility, pressure to produce, and formalization.

Typical statements included "My boss is good at understanding people's problems", "My boss shows trust in the people he manages", "My boss is friendly and approachable", and "It is very important to follow the rules in my organization".

3.2.2. Social Capital and Organizational Silence

The social capital parameter was based on Nahapiet and Ghoshal's [66] research, in which they modified items to measure social capital, with a total of nine items measured on a 5-point Likert scale.

Typical statements included "I have a very good network of colleagues (people around me)", "I usually have a very close relationship with my colleagues (people around me)", "My colleagues and I agree on what is important to the organization", and "My colleagues and I share a vision for the organization".

Organizational silence is a phenomenon in which members intentionally avoid expressing their opinions, thoughts, information, or ideas to advance an organization. In addition to these seven items, Van Dyne, Ang, and Botero [67] provided a framework for this questionnaire. The items included "I do not speak up when I have a good idea because I want to leave the organization", "I do not tell others about solutions to problems that concern me", and "I sometimes deliberately withhold information to protect myself".

3.2.3. Innovative Behavior and Service Performance

The outcome variable, innovation vibrancy, was operationally defined as "the behavior of adopting, spreading, and implementing new ideas". Scott and Bruce's [68] innovative behavior questionnaire was modified to include three items: "I find creative ways to do my job", "I communicate and advocate for new ideas to others", and "I try to obtain the resources I need to implement new ideas".

The act of serving and helping customers is operationalized as employee service performance. The questionnaire developed by Liao and Chuang [69] was modified to suit the hotel context. Representative statements included "I am friendly and helpful to customers" and "I am responsive to customer needs", with six statements measured on a 5-point Likert scale.

**4. Results**

*4.1. Sample Characteristics*

The respondents' demographic characteristics are shown in Table 1. It was found that females outnumbered males. Regarding age, the respondents aged 26–30 years constituted the highest proportion in Korea, while those aged 25 years or younger constituted the highest proportion in China. In terms of educational background, Korea had the most college graduates, while China had the most junior college graduates. In terms of working period, less than five years was the most common in both Korea and China. In terms of position, staff was found to be the most common in Korea, whereas contract workers were the most common in China. In terms of job type, planning/advertising was the most common in both Korea and China.

*4.2. Reliability and Validation*

The research model was validated using a confirmatory factor analysis (CFA). The factor loadings for each item were all valid, and the construct reliability (CR) and average

variance extracted (AVE) for each factor were both higher than the baseline (CR = 0.70, AVE = 0.50). To examine the reliability of the measures, we calculated Cronbach's Alpha Coefficients and found them reliable at 0.70 or higher (Table 2).

**Table 2.** Reliability and validation.

| Variables | Items | Factor Loading | | Cronbach's $\alpha$ | | CR | | AVE | | Model Fit |
|---|---|---|---|---|---|---|---|---|---|---|
| | | Korea | China | Korea | China | Korea | China | Korea | China | |
| Supervisory Support | ss1 | 0.759 | 0.745 | 0.880 | 0.849 | 0.916 | 0.884 | 0.732 | 0.656 | |
| | ss2 | 0.835 | 0.775 | | | | | | | |
| | ss3 | 0.861 | 0.767 | | | | | | | |
| | ss4 | 0.765 | 0.770 | | | | | | | |
| Formalization | f1 | 0.783 | 0.782 | 0.809 | 0.791 | 0.864 | 0.848 | 0.680 | 0.651 | |
| | f2 | 0.781 | 0.754 | | | | | | | |
| | f3 | 0.737 | 0.710 | | | | | | | |
| Pressure to Produce | ptp1 | 0.842 | 0.706 | 0.853 | 0.782 | 0.889 | 0.840 | 0.729 | 0.636 | |
| | ptp2 | 0.864 | 0.752 | | | | | | | |
| | ptp3 | 0.739 | 0.754 | | | | | | | |
| Organizational Silence | as1 | 0.792 | 0.795 | 0.933 | 0.908 | 0.940 | 0.871 | 0.690 | 0.500 | |
| | as2 | 0.827 | 0.851 | | | | | | | |
| | as3 | 0.829 | 0.837 | | | | | | | |
| | as4 | 0.861 | 0.816 | | | | | | | |
| | ds1 | 0.825 | 0.727 | | | | | | | |
| | ds2 | 0.773 | 0.683 | | | | | | | $\chi^2 = 3721.261$ df = 1617 |
| | ds3 | 0.810 | 0.659 | | | | | | | $p = 0.000$ CFI = 0.930 |
| | sc1 | 0.706 | 0.626 | | | | | | | GFI = 0.880 AGFI = 0.860 NFI = 0.884 |
| | sc2 | 0.699 | 0.628 | | | | | | | RMR = 0.030 RMSEA = 0.029 |
| | sc3 | 0.754 | 0.586 | | | | | | | |
| Social Capital | rc1 | 0.713 | 0.702 | 0.915 | 0.881 | 0.944 | 0.907 | 0.650 | 0.521 | |
| | rc2 | 0.812 | 0.752 | | | | | | | |
| | rc3 | 0.738 | 0.671 | | | | | | | |
| | cc1 | 0.723 | 0.666 | | | | | | | |
| | cc2 | 0.742 | 0.728 | | | | | | | |
| | cc3 | 0.756 | 0.688 | | | | | | | |
| Innovative Behavior | ib1 | 0.783 | 0.710 | 0.838 | 0.751 | 0.871 | 0.814 | 0.692 | 0.593 | |
| | ib2 | 0.818 | 0.694 | | | | | | | |
| | ib3 | 0.788 | 0.717 | | | | | | | |
| Service Performance | sp1 | 0.768 | 0.724 | 0.889 | 0.855 | 0.935 | 0.906 | 0.707 | 0.616 | |
| | sp2 | 0.793 | 0.657 | | | | | | | |
| | sp3 | 0.794 | 0.693 | | | | | | | |
| | sp4 | 0.715 | 0.713 | | | | | | | |
| | sp5 | 0.755 | 0.704 | | | | | | | |
| | sp6 | 0.699 | 0.734 | | | | | | | |

Additionally, the model's goodness of fit was evaluated by focusing on goodness-of-fit indicators, such as $\chi^2$, RMR, NFI, CFI, and RMSEA. The measurement model's fit, comprising seven factors, met the standards (Table 2).

This study collected data through a survey at one point in time; thus, common method bias was possible. Harman's [70] single-factor test method was used to determine the existence of this problem. Specifically, it was assessed that the same-method convenience problem did not have a significant impact on the analysis because the distributed explanatory power of one fixed factor and a nonrotational method was 35.583, which was less than half of the total explanatory power.

### 4.3. Correlation Analysis

Before testing the hypotheses, a correlation analysis was conducted to determine the relationships between the variables. The results showed that the independent variables, dependent variables, and parameters were all correlated (Table 3). Additionally, if the correlation between the variables was higher than 0.80, there was a possibility of multicollinearity. However, the correlation between each variable in this study was less than 0.80; thus, multicollinearity was not a concern.

**Table 3.** Correlation analysis.

|  |  | 1 | 2 | 3 | 4 | 5 | 6 | 7 |
|---|---|---|---|---|---|---|---|---|
| Korea | 1. Supervisory Support | 1 |  |  |  |  |  |  |
|  | 2. Formalization | −0.481 ** | 1 |  |  |  |  |  |
|  | 3. Pressure to Produce | 0.570 ** | −0.653 ** | 1 |  |  |  |  |
|  | 4. Organizational Silence | −0.351 ** | 0.331 ** | −0.382 ** | 1 |  |  |  |
|  | 5. Social Capital | 0.705 ** | −0.536 ** | 0.556 ** | −0.332 ** | 1 |  |  |
|  | 6. Innovative Behavior | 0.532 ** | −0.453 ** | 0.567 ** | −0.257 ** | 0.556 ** | 1 |  |
|  | 7. Service Performance | 0.501 ** | −0.677 ** | 0.556 ** | −0.356 ** | 0.553 ** | 0.468 ** | 1 |
| China | 1. Supervisory Support | 1 |  |  |  |  |  |  |
|  | 2. Formalization | −0.487 ** | 1 |  |  |  |  |  |
|  | 3. Pressure to Produce | 0.473 ** | −0.655 ** | 1 |  |  |  |  |
|  | 4. Organizational Silence | −0.165 ** | 0.131 ** | −0.096 ** | 1 |  |  |  |
|  | 5. Social Capital | 0.674 ** | −0.501 ** | 0.539 ** | −0.157 ** | 1 |  |  |
|  | 6. Innovative Behavior | 0.468 ** | −0.541 ** | 0.650 ** | −00.038 | 0.532 ** | 1 |  |
|  | 7. Service Performance | 0.435 ** | −0.668 ** | 0.641 ** | −0.129 ** | 0.525 ** | 0.566 ** | 1 |

Note: ** $p < 0.01$.

### 4.4. Hypothesis Testing

To test the hypotheses, the direct effects of potential variables using a structural equation model were examined, and the following results were obtained: Among the organizational climate variables, supervisory support was positively related to social capital ($\beta = 0.608$, $p < 0.001$), and pressure to produce was significantly related to social capital ($\beta = 0.313$, $p < 0.001$). Among organizational climate, formalization was positively related to organizational silence ($\beta = 0.235$, $p < 0.001$). Social capital was positively related to service performance ($\beta = 0.644$, $p < 0.001$), and organizational silence was negatively related to innovative behavior ($\beta = -0.107$, $p < 0.05$). Therefore, hypotheses 1a, 1b, 2, 3, and 4 were supported (Figure 2).

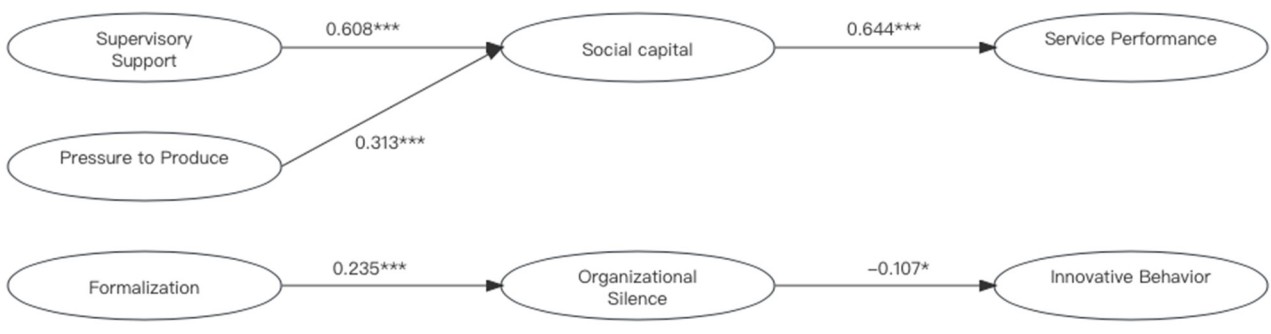

NOTE: ***p<0.001, *p<0.05

**Figure 2.** Direct effect analysis results.

To verify the indirect effects of organizational climate, service performance, and innovative behavior, 2000 bootstrap analyses were conducted, and the following results were obtained: The indirect effect between supervisory support and service performance was 0.293 **, that between pressure to produce and service performance was 0.143 **, and that between formalization and innovative behavior was −0.015 *. Thus, social capital was found to mediate the relationships among supervisory support, pressure to produce, and service performance, while organizational silence was found to mediate the relationship between a formalized organizational climate and innovative behavior. Therefore, hypotheses 5 and 6 were supported. The Korean employee group showed an indirect effect of 0.317 between supervisory support and service performance, of 0.127 between pressure to produce and service performance, and of −0.085 between formalization and innovative behavior. The Chinese employee group found an indirect effect of 0.283 between supervisory support

and service performance, of 0.149 between pressure to produce and service performance, and of −0.005 between formalization and innovative behavior (Table 4).

**Table 4.** Indirect effect analysis results.

| Independent Variable | Parameters | Dependent Variable | Total | Korea | China |
|---|---|---|---|---|---|
| Supervisory Support | Social capital | Service Performance | 0.293 ** | 0.317 ** | 0.283 ** |
| Pressure to Produce | Social capital | Service Performance | 0.143 ** | 0.127 ** | 0.149 ** |
| Formalization | Organizational Silence | Innovative Behavior | −0.015 * | −0.085 ** | −0.005 |

Note: ** $p < 0.01$ and * $p < 0.05$.

According to the results of the country moderation test, organizational climate and social capital did not moderate each other. In the following formats, organizational climate and silence were found to have moderating effects (Figure 3). First, the difference test for the measurement tools in the Korean and Chinese groups indicated no significant differences ($p > 0.5$). Thus, both Koreans and Chinese understood the questions in the questionnaire. Considering the coefficients for the following two groups, formal organizational climate had a positive relationship with organizational silence in both the Korean (ß = 0.436, $p < 0.001$) and Chinese (ß = 0.166, $p < 0.01$) groups. Finally, a significant difference was found in the coefficients for the two groups when the t-value was greater than 1.96. Therefore, hypotheses 7a and 7b were rejected, whereas hypothesis 8 was supported (Table 5).

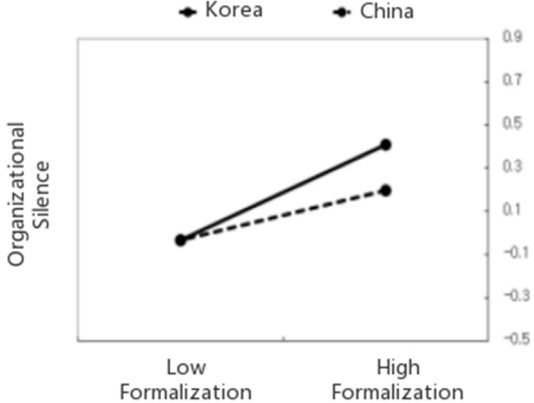

**Figure 3.** Cross-national differences in the relationship between formalization and organizational silence.

**Table 5.** Differences between Korea and China.

| | | | Korea | | China | | t-Value |
|---|---|---|---|---|---|---|---|
| | | | β | *p* | β | *p* | |
| Social Capital | ← | Supervisory Support | 0.630 | *** | 0.596 | *** | −0.843 |
| Social Capital | ← | Pressure to Produce | 0.248 | *** | 0.346 | *** | 1.84 |
| Organizational Silence | ← | Formalization | 0.436 | *** | 0.166 | *** | −2.123 * |
| Innovative Behavior | ← | Organizational Silence | −0.303 | *** | −0.052 | 0.34 | 3.625 ** |
| Service Performance | ← | Social Capital | 0.641 | *** | 0.651 | *** | 0.401 |

Note: *** $p < 0.001$, ** $p < 0.01$, and * $p < 0.05$.

## 5. Discussion

This study examined the effects of organizational climate on employee service performance and innovative behavior, the mediating effects of social capital and organizational silence, and the differences between employees in Korean and Chinese cultures.

First, organizational climate, supervisory support, and pressure to produce were positively related to employee service performance, and formalization was positively related to employee organizational silence. Similar results were obtained for Korean and Chinese employees. An improvement in service quality positively impacts organizational performance [71].

Second, employee service performance positively correlated with social capital, whereas their innovative behavior negatively correlated with organizational silence. When the Korean and Chinese employee groups were separated, social capital was positively correlated with service performance in both groups. However, organizational silence was negatively correlated only for Korean employees; no relationship was found between organizational silence and service performance for Chinese employees. Chinese values are influenced by the great Chinese thinkers [72].

Third, social capital mediated the relationships among supervisor-supportive organizational climate, pressure to produce, organizational climate, and service performance, while organizational silence mediated the relationship between a formalized organizational climate and innovative behavior. Social capital is an essential concept in social science [73]. This study found that organizational silence had no mediating effect on the relationship between a formalized organizational climate and innovative behavior in the Chinese and Korean samples.

### 5.1. Theoretical Implications

The results demonstrated the reliability of the relationships between organizational climate and service quality in different countries. Supervisory support and productive pressure had positive effects on social capital. The social capital perceived by employees increased with the level of organizational support. Appropriate pressure to produce could improve employees' sense of tension and focus on work, thus increasing communication between employees. This can provide guidance for an organization's development.

Chinese people are controlled by the values of Legalism and Taoism, which emphasize systematic control through regulations and laws. Taoism emphasizes that the virtues of adults are so great that the heavens and earth are well-governed by themselves, even if they do nothing [74], suggesting that organizational silence is not simply an antecedent of innovative behavior. Organizational silence negatively impacted Korean enterprises' innovation behavior. The research showed that enterprises' development must create a positive voice and organizational environment, reduce the silence of organizations' members, and encourage employees to actively express their opinions.

The positive relationship between a formalized organizational climate and organizational silence was stronger among Korean employees than among Chinese employees. However, there was no difference between Korean and Chinese employees in terms of the relationships among supervisor-supportive organizational climate, pressure to produce, organizational climate, and social capital. Some immersive management practices can be utilized regardless of cultural values, such as employee education and training [75]. Training is the provision of educational opportunities to improve job performers' knowledge, skills, and abilities [76] and can be utilized without much cultural influence, as it meets the technical requirements of various jobs. The importance and effectiveness of on-the-job training have been consistently emphasized in eastern cultures (e.g., Ng and Norihiko, 2004). Training programs for employees should be designed and delivered considering their preferred ways of learning [77].

This study verified and supported the development of SET and social interaction theory based on a comparison of employee behavior in Korea and China. Based on SET, leadership authorization is the recognition of employees, which is conducive to the estab-

lishment of a harmonious social exchange relationship between employees and leaders, making employees feel obligated to return the care of leaders and, thus, stimulating their innovative behaviors. The choice of a leader is crucial to reducing the cost of innovation [78]. According to social interaction theory, employee development is closely related to their organization, and reasonable communication and leadership strategies can promote enterprises' development. Therefore, SET plays an essential guiding role in organizations' development. Rational leadership and communication strategies can build harmonious social exchange relationships within an organization, thus influencing the organizational atmosphere to move in an optimistic and positive direction, and, through social capital intervention, positively affect employee innovative behavior. Organizational silence reflects the establishment of inadequate social exchange relations. Businesses should avoid this phenomenon and implement timely measures. The results provided more empirical evidence for SET and social interaction theory.

### 5.2. Practical Implications

There are many practical implications of this research. First, organizations should pay attention to supervisory support and pressure to produce in their organizational climates to improve employees' service performances, as well as implementing formalization to improve employees' innovative behaviors.

Second, organizational climate generally enhanced employees' service performances because it increased employee social capital [79]. Organizations can predict employees' performances by observing changes in their social capital. However, organizational climate could inhibit employees' innovative behaviors [80] owing to organizational silence. Thus, organizations may be able to predict their employees' innovative behaviors based on their observations of organizational silence. However, as Chinese employees were influenced by traditional ideas, organizational silence did not predict their innovative behavior.

Third, Korean employees were more likely than Chinese employees to exhibit organizational silence when their organizational climate was formalized. Research confirms that different leadership styles produce different organizational climates, which affect work performance [81], OCBs, and service quality [82]. This suggests that, if an organization has many Korean employees, an overly formalized organizational culture may negatively affect their innovative behavior. In contrast, Chinese employees were less affected by a formalized organizational climate due to the influence of traditional ideas.

This study found that organizational silence had different adverse effects on enterprises' sustainable development in both countries, and extra attention should be paid to this situation in organizational development. Faced with fierce market competition, companies seeking to achieve sustainable development must build a positive organizational climate and stimulate employees' innovative behaviors at work to enhance their competitiveness.

### 5.3. Limitations and Future Research

This study has the following limitations: First, all the variables were similarly measured and may have been influenced by confounding factors. Second, this study was based on employees in South Korea and China; the results may vary when applied to employees in other countries. Follow-up studies can focus on the impacts of digital capabilities and cooperative competitive strategies on firm, sustainable performance [83]. Third, cultural value was not examined in this study. Cultural value is an important aspect of comparative management, as it affects many aspects of individuals and organizations. Future research should try to expand the sample size to include employees from different countries, increase the variable measurement methods, and examine the influence of cultural values.

**Author Contributions:** X.Q. was responsible for methodology and writing—original draft preparation. M.-C.C. was responsible for conceptualization, supervision, and writing—review and editing. X.T. was responsible for formal analysis, investigation, and resources. All authors have read and agreed to the published version of the manuscript.

**Funding:** This work was supported by Jungseok Logistics Foundation.

**Institutional Review Board Statement:** Survey studies in business administration are exempt from review and approval at Gachon University. Ethical review and approval were waived for this study, due to school policy.

**Informed Consent Statement:** Informed consent was obtained from all subjects involved in the study.

**Data Availability Statement:** Not applicable.

**Conflicts of Interest:** The authors declare no conflict of interest.

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
