# Peer review of "Relationship between Organizational Climate and Service Performance in South Korea and China"

_sustainability, doi:10.3390/su151410784_

Round 1
Reviewer 1 Report
Dear Authors,
The paper is interesting but I have some remarks.
1- The abstract seems to be too long, the results must be summarized in the abstract. L.2 "especially in South Korea and China" unnecessary to precise as it is important for all countries.
2- Introduction third paragraph must be rearranged to be more comprehensible and fluent.
3- Fourth paragraph SC and Org.Silence can be covered separately in two paragraphs
4- Moderate English errors and typos and the titles must be checked.
5- Methodology is well chosen and implemented.
6- Theoretical implications can be enriched with support from literature.
7- A short Conclusions part must finish the paper
8- The link with the sustainability issues must be highlighted in the text.
Overall it's a good paper, Best Regards,
- Moderate English errors and typos and the titles must be checked.
Reviewer 2 Report
This research is interesting. However, some improvements are still needed to improve research. In the abstract, it is necessary to state the data collection method, the number of research samples, and research contributions. In the introduction, it is necessary to show the exciting reasons for this research to be carried out and the novelty of the research. In the Theory and hypothesis development section, it is necessary to add a special sub-related to the theory used as the basis for developing hypotheses. The hypothesis development section must be supported by a grand theory to strengthen the hypothesis. In each table, for example, Table 1. For demographic characteristics of the sample, it is necessary to provide source information at the bottom. It is advisable to make a variable operational definition table so that the indicator information for each variable and its source can be identified more clearly. In the discussion section, additional references are needed to support the research results.Author Response
Please see the attachment

Reviewer 3 Report
Dear Author(s),
Thank you for submitting your research paper. I appreciate the clear research question and methodology used in your study. The analysis of the effect of organizational climate on employee innovative behavior and service performance is interesting, especially with the mediating effects of social capital and organizational silence. However, I have a few comments that I believe would help improve the clarity and significance of your findings.
Firstly, it would be helpful if you could provide more detailed information on the cultural, economic, and environmental factors that influence employee behavior in South Korea and China. This would help readers to better understand the context of your study and the implications of your findings.
Secondly, the methodology used to compare the differential effectiveness of organizational climate according to competing values models could be explained more clearly. It would be helpful to provide a detailed explanation of how you divided the three aspects of supervisory support, pressure to produce, and formalization into groups based on human relations, rational goals, and internal process approaches.
Thirdly, the results of your analysis are interesting, but it would be helpful to provide more detailed explanations of the magnitude of the positive effects of supervisory support and pressure to produce on social capital across organizational climates. Additionally, it would be beneficial to explain why Korean employees are more likely to benefit from formalization than Chinese employees in terms of organizational silence.
Lastly, I suggest that you provide a more detailed discussion on the implications of your findings for organizations in South Korea and China. Specifically, how can organizations leverage the positive effects of supervisory support and pressure to produce on social capital to improve employee service performance? How can organizations address the negative effect of organizational silence on innovative behavior among Korean employees?
Overall, your research paper is well-written and structured and provides valuable insights into the effect of organizational climate on employee behavior in South Korea and China. Thank you for the opportunity to review your paper.
Please proofread the manuscript
Reviewer 4 Report
Dear Editor,
Thank you for the opportunity to review the paper. The topic is interesting, but I have few suggestions to the author:
· The abstract should present about research objectives, methodology, and results. It requires improvement.
· Theoretical gaps requires strengthening.
· The theory that supports the hypotheses of the study is not presented in the paper.
· The projected hypotheses lack to the support from similar previous studies.
· Methodology section is mentioned twice in the paper. This section also requires more details about data collection procedure.
· Sampling technique is not highlighted in the paper. Did you use random sampling or nonrandom sampling?
· Common method bias has not been addressed in the paper.
· Validity tests for measurement scales are not conducted. It is important to test convergent and validity assumptions.
· Discussion section should be revised and all hypotheses to be supported by the literature.
· Theoretical contributions should be revised to address the contributions made in this study to the theory and literature.
· There are many grammatical errors in the paper. Author(s) should do proofreading.
Dear Editor,
Thank you for the opportunity to review the paper. The topic is interesting, but I have few suggestions to the author:
· The abstract should present about research objectives, methodology, and results. It requires improvement.
· Theoretical gaps requires strengthening.
· The theory that supports the hypotheses of the study is not presented in the paper.
· The projected hypotheses lack to the support from similar previous studies.
· Methodology section is mentioned twice in the paper. This section also requires more details about data collection procedure.
· Sampling technique is not highlighted in the paper. Did you use random sampling or nonrandom sampling?
· Common method bias has not been addressed in the paper.
· Validity tests for measurement scales are not conducted. It is important to test convergent and validity assumptions.
· Discussion section should be revised and all hypotheses to be supported by the literature.
· Theoretical contributions should be revised to address the contributions made in this study to the theory and literature.
· There are many grammatical errors in the paper. Author(s) should do proofreading.
Round 2
Reviewer 1 Report
I think that your paper can now be published after final reading.
The quality of English seems to be good. Not important issues, after a final reading it will be good to publish.
Author Response
Thank you for your suggestions on the thesis. I read the paper several times and revised it further. In order to make a good thesis, I received English supervision one more time, and I received a total of two English supervision.

Reviewer 4 Report
· The abstract should be one paragraph only. The author should not divide it into two paragraph.
· The sentence “and the data were analysed empirically” mentioned in abstract is not relevant. The author should mention the software used for data analysis.
· The author did not mention research gap as suggested. Talking about social exchange theory and claiming it is a gap is not really convincing. The author(s) should present at the end of introduction how does this study differ from those of other studies. Are there limited studies on the linkages between selected variables?
· Hypotheses still need support from past studies. Any other scholars who found similar results as hypothesized.
· Table 1 “Demographic characteristics of the sample” should be presented in results’ section. The data in the table should be interpreted in a paragraph. It is not enough to present them in a table.
· Figure 2 is too small.
· Discussion section should be separated from theoretical implications and practical implications. Or the author can add theoretical and practical implications provided that discussion should have few paragraph to discuss the research objectives, results, and comparison with those of past studies.
· In discussion section, the author should compare the results with those of past studies.
· Limitations should be in one paragraph instead of splitting them into three
The paper requires proofreading
Author Response

(The authors gave the same response as above.)

Round 3
Reviewer 4 Report
Please combine some of the short paragraphs in introduction and pay attention to the flow between sentences. It is not advisable to keep many short paragraphs (3-5 line only)
Author Response
The introduction has been revised according to the review comments, short paragraphs have been combined and the language expression has been optimized. The introduction section currently has four paragraphs.
